# Starchy staples production shortfalls in Ghana: Technical inefficiency effects outweigh technological differences across ecologies

**Isaac Gershon Kodwo Ansah**[1]\*, **Mark Appiah-Twumasi**[2], **Francis Tsiboe**[3]

**1** Department of Economics, School of Economics, University for Development Studies, Tamale, Ghana,
**2** Department of Agricultural and Food Economics, Faculty of Agriculture, Food and Consumer Sciences, University for Development Studies, Tamale, Ghana, **3** USDA, Economic Research Service, Kansas City, MO, United States of America

\* agershon@uds.edu.gh

**Data Availability Statement:** The data underlying the results presented in the study are available from Ghana Statistical Service and can be accessed freely online via https://www.statsghana.

## Abstract

Starchy staples are a major source of livelihood support for farmers, traders, and processors who participate in these crops' value chains, while also providing staple food to many people, especially the less affluent in society. Despite this position, the productivity figures of starchy staples are low. We use a unique data set and meta-frontier efficiency analysis to assess whether the production shortfalls of major starchy staple crops in Ghana could be attributed to technical inefficiency, technology gaps or both. Results show strong evidence of about 50% production shortfall for cassava, yam, cocoyam, and plantain. For cassava production, the Guinea Savannah zone has the most superior technology, with a technology gap ratio of 0.92, while yam production is more technically efficient in the Sudan Savannah zone, with a technical efficiency score of 0.67. Cocoyam production is more technically efficient (0.56) in the Transition zone, but yam is more technically efficiently produced in the Coastal Savannah zone of Ghana. These results show that production shortfall is more influenced by pure farmer technical inefficiencies (about 45%) rather than by technology gaps (about 20%) along ecological lines. Thus, the sector could benefit from improvements in farmer managerial skills and efficient use of existing technologies.

## 1. Introduction

In 2022, the Food and Agriculture Organization's document on the State of Food Security and Nutrition in the World projected that between 702 and 828 million people faced hunger in 2021, with Africa and Asia being the main hosts of these challenges [1]. In Ghana, a report by GSS/MOFA/WFP/FAO [2] indicate that 11.7% of Ghana's population (3.6 million people) were food insecure in 2020. Out of these, 5.2% were severely food insecure, and 6.5% were moderately food insecure; also, 78% of the food insecure were in rural areas. The report further notes that there are regional heterogeneities in the distribution of food insecurity in Ghana, with most food insecure people living in the Guinea Savannah and Deciduous Forest zones. In explaining the possible causes of these rising food insecurity challenges, the FAO [1] report

gov.gh/gssdatadownloadspage.php. Replication materials are available on GitHub at https://github.com/ftsiboe/Agricultural-Productivity-in-Ghana.

**Funding:** The authors received no specific funding for this work.

**Competing interests:** The authors have declared that no competing interests exist.

identified poor access to food due to low domestic production, poor distribution and high cost of imports, among other factors. Since the primary source of food in these areas, Africa especially, is domestic production, this creates the urgent need to examine the pathways through which food is made available and accessible locally to the vulnerable segment of the population.

In many parts of the developing world, smallholder farmers mostly organize the production of staples. The 2022 FAO report on "The State of Food and Agriculture" point to low agricultural productivity as characterizing smallholder production systems [3]. Based on global projections, Pradhan et al. [4] conclude that Africa has the highest crops yield gaps, and that closing these gaps has the potential to improve food and nutrition security in Africa. Thus, productivity growth and resource use efficiency are important causal pathways to increased food production and access for the poor in these areas. Notably, Pradhan et al. [4] attribute the wide yield gaps in crops, especially those cultivated under rainfed, to biophysical factors (e.g., soil quality, agro-climatic, etc.) and socioeconomic factors (e.g., weather-induced yield variability that affect risk attitudes, market access, etc.). Besides direct food availability, agro-industrial development rests so much on the availability of raw materials to supply industrial needs. How could the industrial need be met if what is available cannot satisfy direct consumption demand? Therefore, productivity growth in these crops is crucial to foster rapid economic transformation in the developing world.

The production of cassava (*Manihot esculenta*), cocoyam (*Colocasia esculenta*), plantain (*Musa sapientum*) and yam (*Dioscorea alata*) (henceforth called starchy staples) plays important roles in the Ghanaian economy. Starchy staples are a major source of livelihood support for farmers, traders, and processors who participate in these crops' value chains, while also providing staple food to many people, especially the less affluent in society. Despite this position, the productivity figures of starchy staples are low. There are non-ignorable productivity gaps between actual and potential yields. In 2018 the national average yields of cassava, cocoyam, plantain, and yam were all over 50% lower than their potential yields, which were estimated at 45Mt/ha for cassava, 38Mt/ha for plantain, 52Mt/ha for yam, and 20Mt/ha for cocoyam [5].

These yield gaps ultimately culminate in production shortfalls which tend to vary widely across regions and ecological zones in Ghana. Such variations indicate that the production shortfalls might be location-specific, crop-specific, and farmer-specific. The large production shortfalls generate costs to the country in terms of wasted resources amid scarcity as well as creating environmental challenges. These production shortfalls, therefore, border on the efficiency of resource utilization (e.g., Ladha et al., [6]) which would require attention. Resource use efficiency, in the layman's understanding, means *doing more with less*. A transition from doing less with more (inefficiency of resource utilization) to doing more with less (efficiency of resource utilization) allows the creation of more economic value with less natural, human, and man-made resources, and eventually enhances household welfare. How can this be achieved if we have limited knowledge of what drives these production shortfalls, or their nature and distribution thereof?

Several studies relevant to Ghanaian agriculture have investigated the primary drivers of resource use efficiency for rice [7–12], maize [12–15], legumes [16–18], cocoa [19–22], and vegetables [23], but with little attention on production shortfalls. An important category of crops that has received less attention despite the important role they play in the food security of households is the starchy staples. A few studies like [24] analyzed starchy staples data for only one season, paying no attention to the possibility of heterogeneous technology adoption within Ghana. Like their peers cultivating vegetables [23], cereals [11, 12, 14], and legumes [25], starchy staples farmers could be operating under different technologies constrained by

the ecology and/or enabling environment. According to the literature, failure to account for these technological differences could lead to falsely attributing production shortfalls due to technology gaps to technical inefficiency [26].

In the last two decades, research on productivity and efficiency analyses in Ghana is highly skewed toward cereal crops. In principle, the starchy staples have received little attention, against the backdrop of the critical roles these crops play, and the significant yield gaps observed in practice. Except for a few studies [12, 22, 23, 25, 27], all efficiency studies in Ghana used data based on a single season, and data size is limited to a few villages or at best a region. This data limitation and lack of locational heterogeneity masks important lessons that could emerge from analyzing a more comprehensive and heterogeneous sample. For example, knowledge on whether technology gaps across ecologies or resource use inefficiencies among farmers contribute most to production shortfalls and how these evolve is not yet clear. Such incomplete information might limit scientific progress, and serve as a barrier to societal development, in the sense that policymakers may be inadequately or wrongly informed. Focusing on starchy staples, this study addresses most of these limitations by accommodating multiple seasons and heterogeneity in ecological technology gaps.

Particularly, this study applies Huang et al. [28] method of estimating meta-stochastic-frontier to a sample of 15,420 starchy staple farmers drawn from nine cross-sectional population-based surveys periodically fielded throughout Ghana from 1987 to 2017. The aim is to examine whether production shortfalls are primarily driven by resource use efficiency among farmers or due to technological gaps across ecological zones in Ghana, as well as their nature and distribution thereof. The pooled data has the broadest coverage of starchy staple households across time and space than previous studies conducted on Ghana. As such, it presents a unique opportunity for utilizing the meta-stochastic frontier to ascertain the ecological and temporal dynamics of starchy staple resource use efficiency in Ghana.

Our results show significant production shortfalls in starchy staples in Ghana, which could be attributed more to pure farmer technical inefficiencies rather than technology gaps. The production technologies used by starchy staple farmers are quite similar across ecologies, but some ecological zones perform better than others. The chronic pure farmer technical inefficiency effects suggest that the production shortfalls could be reduced by addressing the causes of inefficiency and improving the efficient use of existing technologies. These results are important because the achievement of Sustainable Development Goals 2.1 (ensuring access to food for all) and 2.2 (ending all forms of malnutrition) would, at least partly, require the realization of Sustainable Development Goal 12, which leverages the efficient use of natural resources to create a circular economic growth [29].

The rest of the paper is structured as follows. The next section presents an overview of the study area and data. In Section 3, we discuss how the Meta-Stochastic Frontier is used to assess shortfalls in starchy staple production in Ghana. Section 4 presents the results while Section 5 discusses the findings. We present concluding remarks in Section 6.

## 2. Research area and data

### 2.1 Study area

Ghana covers a total land area of 24 million hectares of which 20.7% is considered arable, 40.7% under forests, 11.8% under permanent crops, and 26.8% for other uses. Ghana's ecology is classified as Rain Forest, Semi-Deciduous Forest, Coastal Savannah, Transition, Guinea Savannah, and Sudan Savannah zones. The aridity of these ecologies increases from south to north. Farming systems across these ecologies are highly heterogeneous. However, the highest-yielding ecologies across all the starchy staples are Semi-Deciduous Forest and Transition

zones, and the lowest-yielding ones are Sudan Savannah and Rain-Forest zones. Given their well-balanced annual rainfall and modest temperatures, Semi-Deciduous Forest and Transition zones have the optimal conditions to grow starchy staples. Thus, it is not surprising that they also contain most of the cultivated lands allocated to starchy staples as shown in S1 and S2 Tables in S1 File (all tables, figures, and notes with a leading S indicate supplemental materials in the online appendix).

## 2.2 Data sources

The study uses household level data drawn from two sources: (1) the two waves of the Ghana Socioeconomic Panel Survey (GSPS) fielded in 2009/10 and 2014/15 [30]; and (2) all seven Ghana Living Standards Surveys (GLSS) fielded between 1987–2017 and publicly available at the Ghana Statistical Service (GSS) National Data Archive. The surveys were based on nationally representative samples, focusing on the household as the key socio-economic unit, and provides insights into Ghana's living conditions. The GSPS was collected by the Economic Growth Center (EGC) at Yale University and the Institute of Statistical, Social, and Economic Research (ISSER), and the GLSS by GSS. In terms of the raw data collection, both surveys followed a two-stage stratified sampling design, where enumeration areas and households were selected in the first and second stages, respectively. The harmonization details of these datasets to construct farmer-level data are published in Tsiboe [31]. It is worth noting that while the GSPS constitutes a panel, households are resampled for each GLSS round, hence repeated cross-sectional data. Consequently, since a large proportion of the data is from the GLSS (84%), the harmonized data can be viewed as a cross-sectional sample of the population of Ghanaian starchy staple farmers at roughly five-year intervals.

The final sample used in this study was limited to starchy staple farmers drawn from the various surveys, with yield (kg/ha) above the 5th and below the 90th percentiles by survey, crop, and region. The final sample, therefore, consisted of 15,405 farmers from 15,133 households, out of which 11,585, 3,468, 1,834, and 6,226 farmers cultivate cassava, yam, cocoyam, and plantain, respectively. All of Ghana's ecologies are well represented in the final sample, and the dataset captures 14 growing seasons. Nonetheless, due to sparse data, the Rain Forest and Semi-Deciduous Forest zones were combined into the Forest ecological zone. S1 Note in S1 File in the online appendix documents the definition of all the variables used in this study.

## 2.3 Descriptive statistics

Table 1 provides the descriptive statistics of the household level characteristics, showing their temporal, ecological, and crop variation. Except for land and yield, average values were estimated across all ecologies and by crop. In assessing the differences in these variables across ecology and starchy staples, we used linear regression for continuous variables and a probit model for dummies. A trend variable, and a fixed effect for ecology and crop, as well as their interactions, were included in the estimations. Plots of differences in the household-level variables across ecologies are further elaborated in S1a and S1b Fig in S1 File.

From Table 1, an average household contains about five members in adult equivalence (AE) units, consistent with national estimations of 4.4 and 5.1 in the 2010 and 2000 censuses respectively [32]. The dependency ratio, with an average rate of 1.29, measures the extent of pressure on an earning individual in the household. Averagely, the starchy staple farmer in Ghana is a 47-year-old male with five years of formal education and a 61% chance of owning his main farm plot. Regarding starchy staples, we found no significant variation between males and females (1 = female), education, and land ownership; but in terms of ecological zones, we found significant variation in these variables. Age, on the other hand, showed no significant ecological

**Table 1. Summary statistics of starchy staple producing farmers in Ghana (1987–2017).**

| Variable | Mean (SD) | Trend (%) |
|---|---|---|
| Household [a] | | |
| Size (AE) | 4.83 † (2.887) | 0.10 [0.055] |
| Dependency (ratio) | 1.29 (1.561) | 0.16 [0.113] |
| Farmer [b] | | |
| Female (dummy) | 0.29 † (0.454) | -0.18* † [0.031] |
| Age (years) | 46.84 (14.872) | 0.47* [0.029] |
| Education (years) | 5.01 † (4.820) | 1.59* † ‡ [0.103] |
| Land owned (dummy) | 0.61 † (0.489) | -0.03 † [0.033] |
| Land (ha) [b] | | |
| Cassava | 1.17 † (2.753) | -54.18 † [59.567] |
| Yam | 1.57 † (3.655) | -13.88* † [2.141] |
| Cocoyam | 1.55 (3.883) | 10.54 [832.372] |
| Plantain | 1.19 † (2.611) | -7.18 † [105.680] |
| Yield (Mt/ha) [b] | | |
| Cassava | 0.66 † (1.080) | 2.78* † [0.220] |
| Yam | 0.96 (1.540) | -38.47 † [3146.763] |
| Cocoyam | 0.31 † (0.547) | 0.15 [0.487] |
| Plantain | 0.53 † (0.838) | 1.57* † [0.296] |
| Input use [b] | | |
| Planting material (Mt/ha) | 0.06 † ‡ (0.428) | -5.68 † ‡ [20.428] |
| Family labor (AE) | 2.74 † (1.544) | 0.45* [0.054] |
| Hired labor (man-days/ha) | 11.29 ‡ (43.200) | 1.24* † ‡ [0.350] |
| Pesticide (Liter/ha) | 7.83 † (49.505) | -36.04 † ‡ [2059.161] |
| Enabling environment [a] | | |
| Credit (dummy) | 0.20 † ‡ (0.399) | -0.06* ‡ [0.028] |
| Mechanization (dummy) | 0.04 † ‡ (0.201) | -0.21* ‡ [0.015] |
| Extension (dummy) | 0.19 (0.392) | 0.75* † ‡ [0.029] |

* Indicates significance at $p < 0.05$

† and ‡ indicate statistically significant ($p < 0.05$) differences in the variables across ecology and crop, respectively. The differences were determined via a linear regression for continuous variables and a probit model for dummies. A trend variable, and a fixed effect for ecology and crop, as well as their interactions, were included in the estimation.

[a] Household sample size; Cassava [11,459], Yam [3,439], Cocoyam [1,830], Plantain [6,166], Pooled [15,133]

[b] Farmer sample size; Cassava [11,585], Yam [3,468], Cocoyam [1,834], Plantain [6,226], Pooled [15,405]

Data Sources: Ghana Living Standards Surveys [wave 1–7] and Ghana Socioeconomic Panel Survey [wave 1–2]

or crop variation. In terms of their temporal dynamics from 1987 to 2017, while the probability of observing a female starchy staple farmer significantly declined by 0.18% annually, that of age and years of formal education increased by 0.47% and 1.59% annually, respectively.

In terms of production scale, the value of mean land area confirms the widely held notion that agriculture in Ghana is operated on a smallholder basis. Generally, yam (1.57 ha) has received the largest plot allocation for the period 1987–2017 while cassava (1.17 ha) received the lowest. Starchy staple farm sizes are statistically different across ecologies and have declined over the period (1987–2017). Given these scales of production, yields were estimated at 0.66, 0.96, 0.31, and 0.53 Mt/ha for cassava, yam, cocoyam, and plantain, respectively. Except for cocoyam, these yields also vary by ecology. Nonetheless, except for yam, all crops show an annual upward trend in yields from 1987 to 2017.

Quantities and types of inputs used in production significantly determine technical efficiency and technological levels. Technical efficiency relates to how much output can be obtained from a given input, such as a worker or a machine, or a specific combination of inputs. In other words, it the effectiveness with which a given input set is converted to produce an output. If the farmer is unsuccessful in converting the inputs to the desired outputs, the farmer is said to be technically inefficient. The extent of inefficiency is measured by how much of the desired output is lost due to improper combination of the outputs. Maximum technical efficiency occurs when output is maximised from a given quantity of inputs. The starchy staple production functions analyzed in this study focused on the technical transformation of land, planting materials, family and hired labor, and pesticides into outputs. The planting material usage rate was estimated at 0.06 Mt/ha, with the Sudan Savannah ecology having the highest and Coastal Savannah having the lowest. On average, three household members provided labor for starchy staple production, and this has significantly increased by about 0.45% annually since 1987. Hired labor is used at a rate of about 11 man-days/ha, which is equivalent to an annual increase of about 1.24% since 1987. The highest usage of hired labor is for farms in the Sudan Savannah zone, followed by Transition, Coastal Savannah, Guinea Savannah, and Forest zones, in that order. Farmers in the sample used about 8 liters of pesticides per hectare. Analyzing pesticide use across ecological zones and time shows consistently low usage in the Coastal Savannah relative to the Sudan Savannah and Forest zones for all crops.

Access to credit, mechanization, and extension (measured as dummies) depicts factors that reflect the enabling environment for production success. Access to these production support services is reported in the literature to have diverse effects on technical inefficiency. The rates of access to credit and extension are estimated at 20% and 19% respectively while that for mechanization is a paltry 4%. Furthermore, Table 1 shows that the rate of access to both credit and mechanization has significantly declined annually by 0.06% and 0.21% for credit and mechanization respectively but increased for extension by 0.75% from 1987 to 2017. We present the heterogenous effects of these variables on technical inefficiency across crops and ecological technology gaps in Section 4.

## 3. Econometric strategy

Several studies have shown that farmers in Ghana operate under spatially differentiated technologies [11, 33] which have been linked to variability in Ghana's agroecology. We present Figures in the online appendix and statistical tests in Table 2 that suggest that production characteristics do vary across ecologies and starchy staples. Thus, it is important to account for these spatial heterogeneities driven by agroecological zones when investigating the production performance of starchy staple farmers operating at different levels relative to the best practice frontier. According to Battese et al. [26], not accounting for these technological differences could lead to false attribution of production shortfalls. In the estimation of our empirical model, we group farmers into agroecological technology groups and apply the MSF following Huang et al. [28] two-step method as outlined below.

For each starchy staple in each ecology, we assumed a homogenous farm production technology, which, when coupled with best management practice, situates farmers at various points along their respective ecology Stochastic Frontier (SF). However, due to technical inefficiency and/or idiosyncratic shocks (i.e., downside production risk), one may observe some farmers below the SF [34]. Furthermore, some farmers may also be observed above the SF solely due to upside production risk.

Previous studies have shown that Ghanaian crop production at the micro-level is best captured by the Cobb-Douglas [8, 10], or Translog [11, 22] production function. However, due to

**Table 2. Hypothesis tests for ecology- and meta-frontier models for starchy staple production in Ghana (1987–2017).**

| | Sample size | Log-likelihood | Cobb-Douglas test | Coelli, (1995) [a][b] skewness test | Gutierrez (2001) [a] LR test | Inefficiency variance $[\sigma_u]$ | Total variance $[\sigma^2 = \sigma_u^2 + \sigma_v^2]$ | Gamma $[\gamma = \sigma_u^2/\sigma^2]$ | Inefficiency function test | Model significance |
|---|---|---|---|---|---|---|---|---|---|---|
| **Cassava** | | | | | | | | | | |
| Guinea Savanah | 801 | -919 | 227.02*** | -6.02* | 11.28*** | 0.87 (0.092) | 1.10 (0.119) | 0.68*** (0.076) | 37.20*** | 710.21*** |
| Transitional Zone | 2,064 | -2,735 | 307.98*** | -3.52* | 13.96*** | 0.94 (0.088) | 1.46 (0.118) | 0.61*** (0.066) | 88.27*** | 1194.47*** |
| Forest Zone | 7,023 | -9,129 | 553.51*** | -5.12* | 19.05*** | 0.77 (0.061) | 1.20 (0.065) | 0.50*** (0.053) | 109.64*** | 3941.36*** |
| Coastal Savanah | 1,697 | -2,154 | 211.58*** | -2.84* | 6.13** | 0.80 (0.114) | 1.21 (0.126) | 0.53*** (0.097) | 403.63*** | 1390.95*** |
| National | 11,585 | -15,347 | 1013.19*** | -9.07* | 66.22*** | 0.89 (0.038) | 1.37 (0.048) | 0.58*** (0.030) | 196.70*** | 6319.90*** |
| Meta-frontier | 11,585 | -166 | 16605.98*** | 4555.61 | - | 0.00 (0.036) | 0.05 (0.001) | 0.00 (0.001) | 563.00*** | 91075.02*** |
| **Yam** | | | | | | | | | | |
| Sudan Savanah | 222 | -233 | 164.54*** | -0.23 | 0.52 | 0.87 (0.402) | 1.12 (0.480) | 0.68** (0.339) | 534.06*** | 500.60*** |
| Guinea Savanah | 1,472 | -1,732 | 400.32*** | -9.80* | 34.53*** | 1.04 (0.064) | 1.34 (0.103) | 0.81*** (0.041) | 25.54* | 2378.83*** |
| Transitional Zone | 943 | -1,288 | 199.15*** | -4.89* | 32.23*** | 1.35 (0.091) | 2.16 (0.195) | 0.84*** (0.042) | 1666.17*** | 1328.45*** |
| Forest Zone | 831 | -1,212 | 170.34*** | -3.31* | 23.13*** | 1.41 (0.115) | 2.51 (0.251) | 0.79*** (0.054) | 48.43*** | 772.00*** |
| National | 3,468 | -4,936 | 515.54*** | -12.13* | 226.81*** | 1.53 (0.040) | 2.67 (0.104) | 0.88*** (0.014) | 213.99*** | 4669.74*** |
| Meta-frontier | 3,468 | -1,994 | 3948.97*** | -197.43* | 742.33*** | 0.72 (0.011) | 0.53 (0.015) | 0.97*** (0.003) | 14048.06*** | 38680.49*** |
| **Cocoyam** | | | | | | | | | | |
| Transitional Zone | 422 | -597 | 99.75*** | -1.84 | 9.11*** | 1.38 (0.171) | 2.40 (0.359) | 0.79*** (0.084) | 23.91** | 322.93*** |
| Forest Zone | 1,412 | -2,046 | 75.45*** | -3.13* | 25.43*** | 1.36 (0.104) | 2.37 (0.212) | 0.78*** (0.053) | 42.48*** | 495.20*** |
| National | 1,834 | -2,721 | 113.04*** | -2.54* | 20.51*** | 1.27 (0.106) | 2.29 (0.195) | 0.71*** (0.060) | 54.73*** | 611.93*** |
| Meta-frontier | 1,834 | -377 | 1587.92*** | 658.26 | - | 0.00 (0.082) | 0.08 (0.003) | 0.00 (0.004) | 890.54*** | 8960.81*** |
| **Plantain** | | | | | | | | | | |
| Transitional Zone | 740 | -985 | 125.50*** | -5.65* | 39.00*** | 1.38 (0.085) | 2.21 (0.195) | 0.86*** (0.035) | 63.15*** | 760.94*** |
| Forest Zone | 4,982 | -6,564 | 357.18*** | -6.48* | 36.32*** | 0.97 (0.057) | 1.46 (0.079) | 0.64*** (0.042) | 46.45*** | 3483.09*** |
| Coastal Savanah | 504 | -576 | 109.18*** | 0.28 | - | 0.03 (0.690) | 0.61 (0.046) | 0.00 (0.064) | 1138.60*** | 390.07*** |
| National | 6,226 | -8,275 | 447.69*** | -9.27* | 78.14*** | 1.07 (0.044) | 1.62 (0.068) | 0.70*** (0.029) | 88.04*** | 4366.28*** |
| Meta-frontier | 6,226 | 1,264 | 9511.74*** | -1939.14* | 1062.47*** | 0.40 (0.006) | 0.19 (0.005) | 0.83*** (0.008) | 544.13*** | 113229.19*** |

Significance levels

* p<0.10

** p<0.05

***p<0.01

[a] Null hypothesis of no one-sided error (i.e., no inefficiency) was tested

[b] Values less than the critical value of -1.96 confirm the rejection of the null hypothesis.

Data Sources: Ghana Living Standards Surveys [wave 1–7] and Ghana Socioeconomic Panel Survey [wave 1–2]

its relative flexibility, this study implements the MSF assuming that $f_i^j(\cdot)$ is Translog. Nonetheless, since Cobb-Douglas is nested within the Translog, the former was tested after estimation and was soundly rejected at p<0.01 for all models (see Table 2). Thus, the Translog specification is used. The SF production function for the $j^{th}$ agro-ecology is specified as:

$$f^j(x_i) = \ln y_{ijt} = \beta_{0r} + \sum_k \beta_{kj} \ln x_{kijt} + \frac{1}{2}\sum_s\sum_k \beta_{skj} log \ln x_{kijt} \ln x_{sijt} + v_{ijt} - u_{ijt} \ u_{ijt}$$

$$\sim N^+\left[0, \exp(\mathbf{w}_{ijt}\boldsymbol{\alpha})\right], v_{ijt} \sim N\left(0, \sigma_v^2\right) \tag{1}$$

where, $y_{ijt}$ is total production (Mt) for the $i^{th}$ farmer in ecology $j$ at time $t$. Each $x_{kijt}$ represents the $k^{th}$ input (i.e., land, planting material, family and hired labor, and pesticides) used by the $i^{th}$ farmer and a trend variable. As Van Nguyen et al. [35] report, outcomes from the SF, especially with efficiency estimates, are severely affected by the distribution assumptions. Therefore, we considered all possible distributions but eventually had to settle on a half-normal distribution (i.e., $u_{ijt} \sim N^+[0, \exp(\mathbf{w}_{ijt}\boldsymbol{\alpha}_j)]$), since the truncated normal and exponential distributions presented issues of convergence. Following Tsiboe et al. [23], $\mathbf{w}_{ijt}$ contained farmer characteristics (age, education, and gender), institutional factors (land ownership, credit, and extension), a trend and a constant term. Given the SF production function for the $j^{th}$ agro-ecology, the "pure farmer technical efficiency" (TE) of the $i^{th}$ farmer is calculated as:

$$TE_i = E[\exp(-u_i)|\hat{\varepsilon}_i| \tag{2}$$

To implement Huang et al's. [28] two-step MSF method, Eq (1) is initially estimated separately for each agro-ecology ($j$), and then the predicted output levels from the agro-ecology frontiers are used as the observation for MSF (pooled stochastic frontiers) in the second step. The conventional one-sided error term ($u_{iM}$) for the MSF serves as the estimate for any technology gaps among the different agro-ecology. The MSF [$f^M(x_i)$] which envelops all agro-ecology-specific stochastic frontiers [$f^j(x_i)$] is specified in Eq (3) as

$$f^M(x_i) = \ln \hat{y}_{ijt} = \beta_{0r} + \sum_k \beta_{kM} \ln x_{kiMt} + \frac{1}{2}\sum_s\sum_k \beta_{skM} log \ln x_{kiMt} \ln x_{siMt} - u_{iMt} \tag{3}$$

Where $u_{iM} \sim N^+(0, \exp(\mathbf{w}_i\boldsymbol{\alpha}_M))$ is strictly positive, implying that $f^j(x_i) \leq f^M(x_i)$. Consequently, the ratio of agro-ecology $j$'s stochastic frontier to the MSF is the technology gap ratio (TGR), represented as

$$TGR_i = \frac{f^j(x_i)}{f^M(x_i)} = e^{-u_{iM}} \leq 1 \tag{4}$$

The TGR depends on the accessibility and adoption level of the available MSF which in turn depends on farm-specific circumstances. The meta-technical efficiency (MTE), a measure of overall performance, represents each farmer's technical efficiency with respect to the meta frontier production technology. MTE can be decomposed into the TE (which is the technical efficiency measured with respect to the zone-specific frontier) and the TGR (which is the difference between the best available technology and the technology set adopted). Accordingly, each farmer's MTE is given by Eq (5) as follows:

$$MTE_i = f^j(x_i)[f^M(x_i)e^{v_i}]^{-1} = TGR_i \times TE_i \tag{5}$$

The MSF is estimated separately for each crop, so we use the physical measure of total output and input (log-transformed) used by the farmer. Thus, all outputs and inputs are not standardized by farm size. The parameters of the ecology- and meta-frontiers were estimated via

maximum likelihood, using the "*frontier*" command in Stata 15. Furthermore, estimation variables were standardized by their sample means by survey and crop. Given the parameters, input elasticities were estimated as the first derivative of the frontiers with respect to that input, evaluated at their means. The production returns to scale (RTS) were then estimated as the sum of the input elasticities. The standard errors for the elasticities were estimated via the delta method. Ecology-specific TE, TGR, and MTE were estimated using Eq (2), (4), and (5), respectively.

## 4. Econometric results

The core objective of this paper is to understand whether production shortfalls existing among the starchy staples in Ghana are driven by technological gaps, technical inefficiency, or both. This results section has four parts. First, we present results on diagnostic tests of the empirical model specification before proceeding to the estimates of inputs elasticities, returns to scale (RTS), and technical change for each crop, ecology, and the meta-frontier in the second part. This is followed by the spatiotemporal trends of TE, TGR, MTE, and technology gaps in the third part. The final part presents results on drivers of technical inefficiency and technology gaps. Estimates of the production functions for each crop (S3-S6 Tables in S1 File), ecological variations in production yield, input, and enabling environment factors, and in starchy staple farmer and household demographic factors in Ghana (S1a and S1b Fig in S1 File) are presented in the online appendix.

### 4.1 Diagnostic tests of the empirical model specification

The tests results for functional form specification, skewed error specification, production variance due to technical inefficiency, and hypothesis tests that variables in the inefficiency functions of ecology-specific and meta-frontier models do not affect starchy staple inefficiency are presented in Table 2.

The Cobb-Douglas and Translog functional forms are the most popular choices when specifying MSF production functions. This study tested the null hypothesis that cross terms in the translog functional form are jointly equal to zero. The test result for all crops, ecologies, and the meta-frontier soundly rejected this hypothesis at 1% level, showing that the translog function better fits the data. In the literature, the translog function is generally preferred because of its flexibility. For example, both Asravor et al. [11] and Tsiboe [19] used the translog specification for the MSF for rice and cocoa production in Ghana.

The results from the Coelli [36] skewness test for negative OLS residuals and the Gutierrez et al. [37] likelihood ratio tests for technical inefficiency both reject the null hypothesis of no one-sided error and show that the MSF modeled for starchy staple technical inefficiency functions is justified. Further, the null hypothesis that the inefficiency variance is not influenced by variables in the inefficiency function ($H_o$: $\boldsymbol{\alpha} = 0$) was also rejected for all crops, ecologies, and the meta-frontier.

The proportion of production variance due to inefficiency ($\gamma$) ranged from 0.50 to 0.99 for the ecology frontiers; for a given crop, this was always higher than that of the meta-frontier.

The $\gamma$ values suggest that a considerable amount of the observed variation in output for the ecology-frontiers could be attributed to inefficient use of inputs. The range of $\gamma$ for the meta-frontier also suggests that the observed variation in output, given the ecology frontiers, could not be attributed to idiosyncrasies.

To statistically check the similarities of the production frontiers for a given starchy staple across the different ecologies, the study applied a generalized likelihood ratio test. The log-likelihood values under the null hypothesis of a uniform technology were all smaller than that of

**Table 3. Elasticities for ecology- and meta-frontier models for starchy staple production in Ghana (1987–2017).**

| | Land elasticity | Planting material elasticity | Family labor elasticity | Hired Labor elasticity | Pesticide elasticity | Returns to scale [a] |
|---|---|---|---|---|---|---|
| Cassava | | | | | | |
| Guinea Savanah | 0.31*** (0.030) | -0.13*** (0.025) | 0.16*** (0.063) | 0.05*** (0.008) | 0.00 (0.014) | 0.39*** (0.069) |
| Transitional Zone | 0.36*** (0.019) | 0.02 (0.018) | 0.10** (0.042) | 0.04*** (0.009) | 0.01 (0.007) | 0.53*** (0.047) |
| Forest Zone | 0.40*** (0.010) | -0.04*** (0.010) | 0.14*** (0.023) | 0.04*** (0.003) | -0.01* (0.004) | 0.53*** (0.026) |
| Coastal Savanah | 0.41*** (0.020) | -0.05*** (0.019) | 0.17*** (0.046) | 0.04*** (0.007) | -0.09*** (0.018) | 0.47*** (0.059) |
| National | 0.39*** (0.008) | -0.03*** (0.008) | 0.16*** (0.018) | 0.04*** (0.003) | -0.01** (0.004) | 0.55*** (0.020) |
| Meta-frontier | 0.39*** (0.002) | -0.04*** (0.002) | 0.16*** (0.005) | 0.04*** (0.001) | -0.01*** (0.001) | 0.54*** (0.006) |
| Yam | | | | | | |
| Sudan Savanah | 0.44*** (0.068) | -0.08** (0.032) | 0.18* (0.103) | 0.01 (0.015) | -0.03 (0.020) | 0.51*** (0.133) |
| Guinea Savanah | 0.59*** (0.025) | -0.08*** (0.019) | 0.02 (0.047) | 0.02*** (0.005) | -0.02 (0.013) | 0.52*** (0.056) |
| Transitional Zone | 0.55*** (0.035) | 0.00 (0.020) | -0.05 (0.061) | 0.10*** (0.016) | 0.02 (0.011) | 0.61*** (0.072) |
| Forest Zone | 0.32*** (0.030) | -0.08* (0.043) | 0.20*** (0.076) | 0.09*** (0.014) | -0.01 (0.014) | 0.53*** (0.091) |
| National | 0.55*** (0.019) | -0.06*** (0.013) | 0.12*** (0.035) | 0.05*** (0.005) | -0.02*** (0.007) | 0.64*** (0.040) |
| Meta-frontier | 0.51*** (0.009) | -0.07*** (0.005) | 0.10*** (0.019) | 0.06*** (0.002) | -0.01*** (0.003) | 0.58*** (0.021) |
| Cocoyam | | | | | | |
| Transitional Zone | 0.19*** (0.040) | -0.18** (0.083) | 0.21** (0.104) | 0.08*** (0.022) | -0.03 (0.034) | 0.27*** (0.134) |
| Forest Zone | 0.42*** (0.041) | -0.10*** (0.037) | 0.08 (0.059) | 0.04*** (0.010) | 0.02 (0.016) | 0.45*** (0.076) |
| National | 0.36*** (0.027) | -0.10*** (0.035) | 0.09* (0.052) | 0.05*** (0.009) | 0.01 (0.014) | 0.42*** (0.066) |
| Meta-frontier | 0.38*** (0.005) | -0.10*** (0.015) | 0.11*** (0.015) | 0.05*** (0.003) | 0.01*** (0.004) | 0.46*** (0.023) |
| Plantain | | | | | | |
| Transitional Zone | 0.44*** (0.031) | -0.04 (0.032) | 0.10 (0.066) | 0.06*** (0.011) | 0.02 (0.015) | 0.59*** (0.073) |
| Forest Zone | 0.47*** (0.014) | -0.03** (0.013) | 0.14*** (0.027) | 0.04*** (0.004) | 0.00 (0.005) | 0.62*** (0.031) |
| Coastal Savanah | 0.37*** (0.032) | 0.06* (0.033) | 0.12* (0.073) | 0.07*** (0.015) | -0.05** (0.023) | 0.58*** (0.098) |
| National | 0.47*** (0.012) | -0.02* (0.011) | 0.14*** (0.025) | 0.04*** (0.004) | 0.00 (0.005) | 0.63*** (0.028) |
| Meta-frontier | 0.47*** (0.002) | -0.03*** (0.002) | 0.14*** (0.006) | 0.04*** (0.001) | 0.01*** (0.001) | 0.63*** (0.007) |

Significance levels

* $p < 0.10$

** $p < 0.05$

***$p < 0.01$

[a] Null hypothesis of constant returns to scale was tested.

Data Sources: Ghana Living Standards Surveys [wave 1–7] and Ghana Socioeconomic Panel Survey [wave 1–2]

the alternate hypothesis of heterogeneous technology. The test statistics led to sound rejection of the null hypothesis, implying that starchy staple farmers are operating under heterogeneous technologies along ecological lines.

Finally, the SF model chi-squared test statistics indicate that the models are all significant. These justify further inquiry into whether technology gaps across ecologies or resource use inefficiencies among farmers contribute more to production shortfalls. We report the summary of estimates for the ecology- and meta-frontier production elasticities in Table 3 and TGR, TE, and MTE for starchy staples production in Table 4. Detailed results from these are discussed in the following subsections along crop-specific lines.

## 4.2 Output elasticities with respect to inputs, returns to scale and technical change

Given that earlier results confirmed heterogeneous production technologies along ecological lines, we estimated separate ecology-frontier and meta-frontier models for each crop and

**Table 4. Ghanaian starchy staple production technology level and technical efficiency across ecologies (1987–2017).**

|  | Technology gap ratio (TGR) | Technical efficiency (TE) | Meta-technical efficiency (MTE) | Technology gap |
|---|---|---|---|---|
| Cassava |  |  |  |  |
| Guinea Savanah | 0.92*** (0.002) | 0.64*** (0.007) | 0.60*** (0.006) | 8.50% |
| Transitional Zone | 0.90*** (0.001) | 0.58*** (0.004) | 0.54*** (0.003) | 9.69% |
| Forest Zone | 0.87*** (0.000) | 0.64*** (0.001) | 0.56*** (0.001) | 12.82% |
| Coastal Savanah | 0.84*** (0.003) | 0.72*** (0.005) | 0.59*** (0.003) | 16.31% |
| Mean | 0.89 | 0.65 | 0.58 | 11.44% |
| Yam |  |  |  |  |
| Sudan Savanah | 0.81*** (0.009) | 0.67*** (0.018) | 0.55*** (0.016) | 18.72% |
| Guinea Savanah | 0.81*** (0.003) | 0.54*** (0.005) | 0.43*** (0.004) | 18.70% |
| Transitional Zone | 0.70*** (0.004) | 0.49*** (0.007) | 0.34*** (0.005) | 30.00% |
| Forest Zone | 0.55*** (0.005) | 0.51*** (0.008) | 0.24*** (0.005) | 45.00% |
| Mean | 0.65 | 0.54 | 0.33 | 35.11% |
| Cocoyam |  |  |  |  |
| Transitional Zone | 0.77*** (0.009) | 0.56*** (0.011) | 0.38*** (0.009) | 22.66% |
| Forest Zone | 0.82*** (0.002) | 0.48*** (0.005) | 0.39*** (0.004) | 17.76% |
| Mean | 0.80 | 0.47 | 0.36 | 20.21% |
| Plantain |  |  |  |  |
| Transitional Zone | 0.94*** (0.002) | 0.50*** (0.008) | 0.47*** (0.007) | 6.39% |
| Forest Zone | 0.87*** (0.000) | 0.58*** (0.002) | 0.51*** (0.002) | 13.01% |
| Coastal Savanah | 0.75*** (0.005) | 0.68*** (0.009) | 0.50*** (0.008) | 25.07% |
| Mean | 0.85 | 0.58 | 0.50 | 15.13% |
| Overall mean | 0.791 | 0.553 | 0.428 | 20.94% |

Significance levels: * p<0.10, ** p<0.05, ***p<0.01

[a] Null hypothesis of constant returns to scale was tested.

Data Sources: Ghana Living Standards Surveys [wave 1–7] and Ghana Socioeconomic Panel Survey [wave 1–2]

estimated local and national returns to scale (RTS) parameters from these models to test the null hypothesis of constant RTS from each model. The estimated elasticities for ecology-specific and meta-frontier models for all starchy staples are presented in this order in Table 3.

**4.2.1 Cassava production.** Cassava is produced in all ecological zones under study except Sudan Savannah. The elasticities for land are consistently statistically significant and contribute the most to explaining changes in cassava production efficiency across all ecologies and the meta-frontier. Land elasticity is highest in the Coastal Savannah zone (0.41) and lowest in the Guinea Savannah (0.31) while the national average land elasticity of 0.39 is like the meta-frontier estimate. Family labor contributes the next highest elasticities followed by hired labor. Again, the estimate for family labor is highest in the Coastal Savana (0.17) but the lowest for Transition zone (0.10). The national and meta-frontier elasticities average about 0.16 each. The estimated elasticities for hired labor are comparable across ecologies and the meta-frontier with the Guinea Savannah zone (0.05) slightly outperforming the rest (0.04). Where statistically significant, the elasticities for planting materials and pesticides reduce both the production frontiers and ecological technologies for cassava production. The negative effect for planting material is highest in the Guinea Savannah zone (-0.13) and is, at least, three times the estimated national (-0.03) and meta-frontier (-0.04) effects. The negative effect of pesticide use on cassava output is even more distressing since the estimated elasticity for Coastal Savannah (-0.09) outstrips the national and meta-frontier elasticities by about nine-folds.

Taken together, cassava production exhibits decreasing RTS for all ecologies and at the national and meta-frontier levels. The ecology RTS for cassava production ranges from 0.39 in the Guinea Savannah to 0.53 in both the Transitional and Forest zones. The national (0.55) and meta-frontier (0.54) RTS for cassava align with the agro-ecology findings.

**4.2.2 Yam production.** From Fig 2 and Table 3, yam is produced in all agro-ecologies except Coastal Savannah. Like cassava, land contributes the most to explaining variations in yam output followed by family labor and hired labor. Pesticides and planting materials maintain their negative effects on output and ecology technology. Guinea Savannah (0.59) shows the most while Coastal Savannah (0.32) shows the least responsiveness to marginal changes in area cultivated. The elasticity of family labor in the Forest zone (0.20) is about 60% higher than the national elasticity (0.12) and twice that of the meta-frontier. Although family labor showed no effect in the Transition zone, hired labor elasticity (0.10) is twice the national average. Where statistically significant, planting material had negative effects on yam output at the ecological and meta-frontier levels. The elasticities of yam output with respect to changes in planting material clustered around -0.08 in the forest (p<0.10), Sudan Savannah (p<0.05), and Guinea Savannah (p<0.01) zones: seed yam showed no effect on output in the Transition zone. Like planting material, pesticide had generally negative effects on yam output (except in the Transition zone) but none of the ecology-specific elasticities were statistically significant. These results reflect the heterogenous production technologies for the same crop in different ecologies. At the meta-frontier level, both planting material (-0.07) and pesticide (-0.01) showed negative effects on yam output. Like cassava production, the RTS for yam also reflects decreasing returns to scale in all ecologies and the meta-frontier although the national level estimate (0.64) is the highest among all crops.

**4.2.3 Cocoyam production.** Cocoyam production in Ghana happens largely in the Transition and Forest ecological zones. These areas have more suitable biological conditions for this crop, but the estimated elasticities show marked heterogeneities in input use and production technologies. For example, land elasticity in the Forest zone (0.42) is more than twice the effect in the Transition zone (0.19); the elasticity of family labor in the Transition zone (0.21) is the highest of all inputs in that zone but family labor shows no effect on cocoyam output in the Forest zone. Hired labor shows complementary relationships with family labor in both ecologies but its effect in the Transition zone is twice (0.08) that of the Forest zone. Pesticide application does not significantly influence cocoyam output while planting material maintains its negative output effects in both ecologies (Transition = -0.18; Forest = -0.10) and in the meta-frontier (-0.10). These negative effects are reflected in the low RTS for the Transition zone (0.27). Cumulatively, the meta-frontier RTS for cocoyam production (0.46) is the least among the starchy staples and implies that a 100% increase in all inputs will only yield a 46% increase in cocoyam output.

**4.2.4 Plantain production.** Plantain is mainly cultivated in the Transition, Forest, and Coastal Savannah zones of Ghana. Its ecology-specific and meta-frontier elasticities show that land contributes most to output, and the effect is more pronounced in the Forest zone (0.47) but lowest in the Coastal Savannah zone (0.37). The estimated land elasticity of 0.47 in the Forest zone is consistent with its analogues at the national and meta-frontier levels. Similar observations can be made for the elasticities of family labor where the estimate for Forest zone (0.14) is the same for the national and meta-frontier level. Interestingly, the elasticities of family labor (0.10) and planting material (-0.04) in the Transition zone have no significant effect on plantain output as was the case for yam production in that zone. Hired labor showed consistently significant and positive effects on plantain output with the elasticities ranging from 0.04 in the Forest zone to 0.07 in the Coastal Savannah while planting material and pesticide elasticities continued their largely negative effects. Interestingly, the only statistically

significant effect of pesticide elasticity (-0.05) was found in the Coastal Savannah zone and, that zone also recorded the only positive effect of planting materials on plantain output. Like the others, plantain production also showed decreasing RTS at the national and meta-frontier levels although the estimates for the latter (0.63, p<0.01) was the highest of all crops under study.

## 4.3 Technical efficiency, technology gap, and meta-technical efficiency of starchy staples

The summary of TE, TGR, and MTE estimates for agro-ecological zones are presented in Table 4, while Figs 1 and 2 present the corresponding plots. As indicated, these efficiency

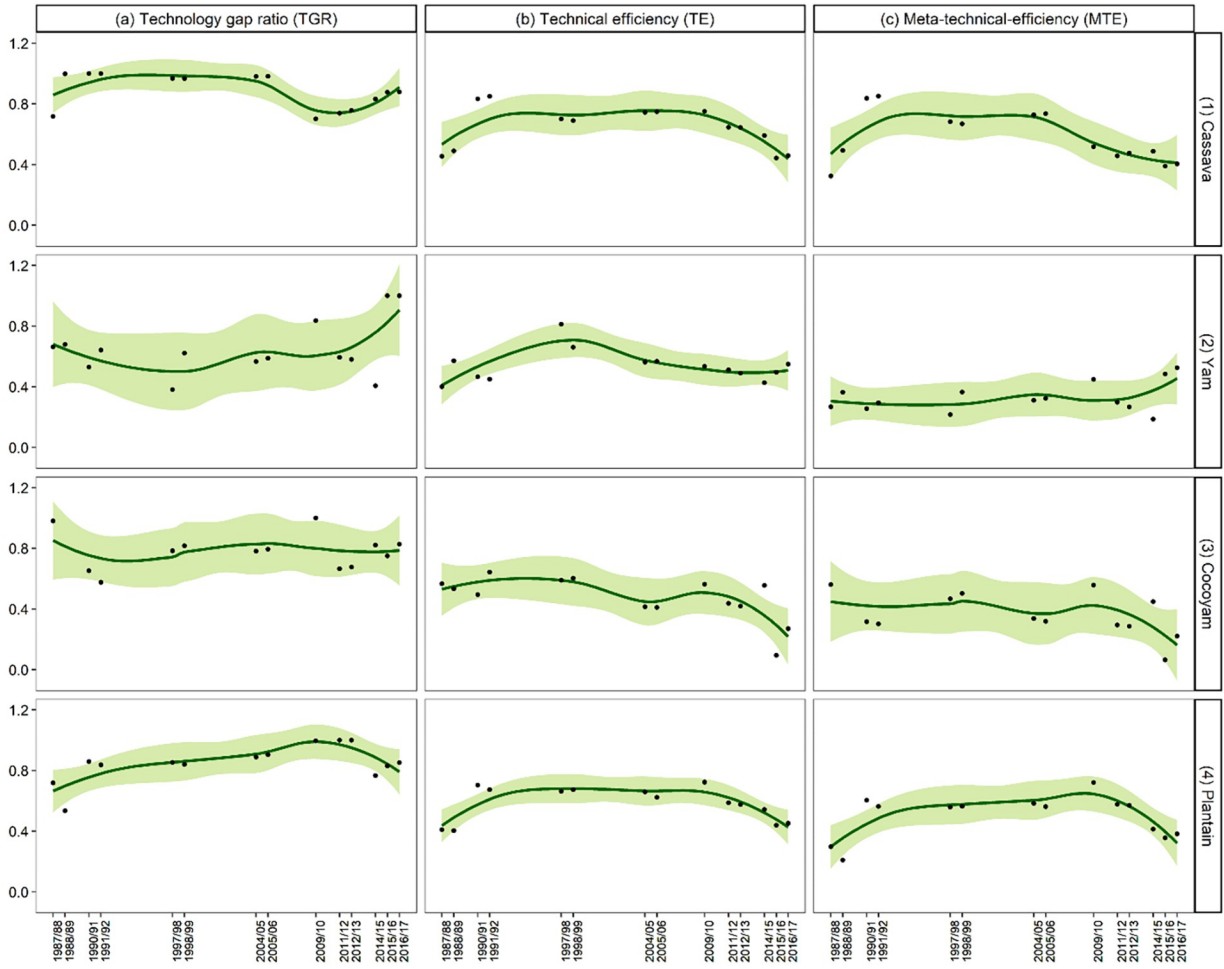

**Fig 1. Ghanaian starchy staple production technology level and technical efficiency parameters across seasons (1987–2017). Notes:** Farmer-level scores were first estimated via a Meta–stochastic–frontier (MSF) analysis applied separately to 9 population–based surveys that represent 30 years of farmer-level data collection in Ghana the surveys used included Ghana Living Standards Surveys [wave 1–7] and Ghana Socioeconomic Panel Survey [wave 1–2]. The farmer-level elasticities were subsequently averaged across seasons via a regression framework to account for controls. Each point on a sub–panel represents the mean of the estimates. Given the seasonal means, the fitted line was done locally using neighborhood points, weighted by distance. The size of the neighborhood was set to 75% of the points with a tri–cubic weighting. The gray region is the 95% confidence interval of the fitted line.

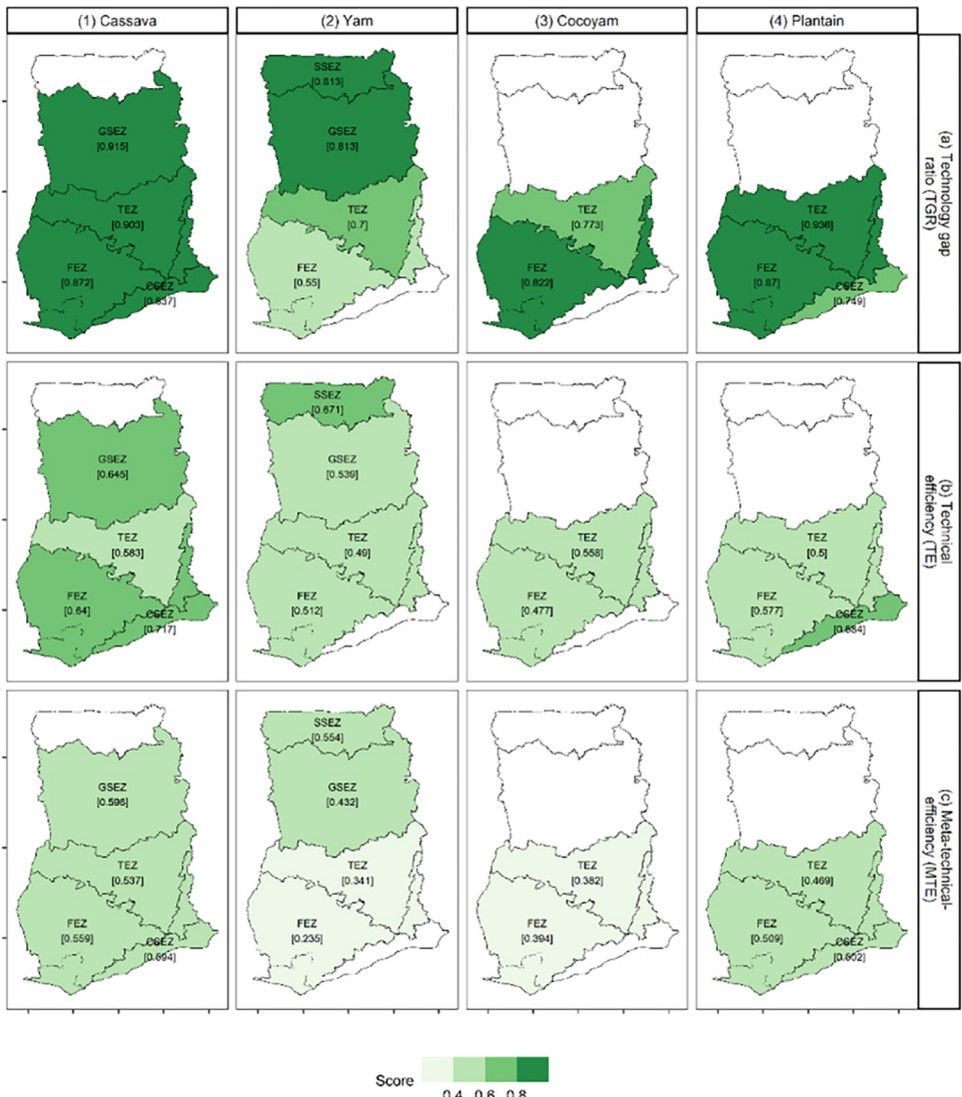

**Fig 2. Ghanaian starchy staple production technology level and technical efficiency across ecologies (1987–2017).**
**Notes:** Farmer-level scores were estimated via a Meta−stochastic−frontier (MSF) analysis applied separately to 10 population−based surveys that represent 30 years of farmer-level data collection in Ghana the surveys used included Ghana Living Standards Surveys [wave 1−7] and Ghana Socioeconomic Panel Survey [wave 1−2].

parameters are estimated from Eq 2, 4, and 5. Detailed results from these estimations are presented for each starchy staple.

**4.3.1 Cassava production.** The technology gap (ranging from 0 to 1) is the difference between the best technology available in the production of a particular crop and the technology set adopted by a farmer. TGR values closer to one mean there is little room for improvement given the existing production technology in the country. The choice of the adopted technology is determined by environmental and socio-economic factors. The mean TGR for cassava, estimated at 0.89, indicates that the technological gap averaged over all ecological zones is 11%. Fig 1, panel a1 shows that the TGR for cassava increased from 0.72 in 1987 to 0.98 in 1998 and dropped to an average of 0.85 between 2004 and 2012. Afterward, it increased until settling at 0.88 in 2017. Given the high mean TGR across the sample, it is not surprising that the ecology

specific TGRs are all above 80% with the highest in Guinea Savannah (0.92), reflecting a superior technology for that ecology. This is followed by the Transition zone (0.90), the Forest zone (0.87), and then Coastal Savannah (0.84). The TGR for cassava is relatively high when compared to 0.75 for tomato [23] and 0.74 for rice [11].

In terms of TE, the mean ecology-specific estimate for cassava is 0.65 across the entire sample. Trend-wise, the TE for cassava increased by about 90% from 0.46 in 1987 to a peak of 0.85 in 1991 but has been declining since then by about 5% annually to 0.46 again in 2017 (Fig 1, panel b1). Fig 2 shows that the mean TE for cassava is highest in Coastal Savannah (0.72) and followed by Forest/Guinea Savannah zones (0.64) and then Transition zone (0.58). An important caveat about these TE scores is that they do not rank the farms based on the cassava metafrontier, so we discuss the MTE results next.

After accounting for ecology-specific production technologies, the mean MTE for cassava across the entire sample is 0.58. Specifically, the most technically efficient cassava farmers relative to the meta-frontier are those in the Guinea Savannah with an MTE of 0.60.

Fig 2 shows that the Guinea Savannah farmers are followed by their peers in Coastal Savannah (0.59), Forest zone (0.56), and then Transition zone (0.54). Fig 1, panel c1 shows that the MTE increased from 0.32 in 1987 to 0.85 in 1991 due to improvements in technology adoption. However, from 1991 to 2017, MTE declined at the same rate as pure farmer technical efficiency to 0.40. We return to the implications and policy learning points from these findings in the Discussion section.

**4.3.2 Yam production.** The mean TGR for yam, estimated at 0.65, points to an average ecological technology gap of about 35%. Fig 1, panel a2 shows that the TGR for yam has increased steadily over the study period but this generalization masks significant dips and peaks over time. For example, the TGR for yam dropped by about 4.25% annually in the decade between 1987 (0.66) and 1997 (0.38); this rebounded to 0.62 in 1998 only to sink to 0.41 in 2014 before hitting another gear shift to peak at 1.00 in both 2016 and 2017. At the ecology level, the highest TGR is estimated at 0.81 for Sudan Savannah/Guinea Savannah, followed by the Transition zone (0.70), and then the Forest zone (0.55).

Ecology-specific TE for yam is estimated at a mean of 0.54 and has remained relatively constant over the study period (Fig 1, panel b2). Yam TE follows similar patterns as the TGR, the only difference being that the estimate for the Forest zone (0.51) is higher than the Transition zone (0.49) (see Fig 2). The TE values for yam estimated in this study are relatively low when compared to values of 0.85 and 0.89 estimated by Owusu Asante et al. [24] for Ashanti and Brong Ahafo regions, respectively (all approximately located in the Transition zone). Several things distinguish this study from Owusu Asante et al. [24] that could account for the differences in TE. First, our study contains production data from multiple regions/ecologies throughout Ghana and multiple production seasons, while Owusu Asante et al. [24] used data from only 375 yam farmers in one ecology and for only one season. Thus, the values estimated by Owusu Asante et al. [24] do not holistically capture the spatiotemporal dynamics of yam TE as this study does. Secondly, Owusu Asante et al. [24] utilized data that is over-represented by those farmers (58%) using yam mini-sett technology which the authors show induces positive and significant effects on the TE of some of its adopters. By the nature of its data, this study is not able to pinpoint the specific technologies used by each farmer. However, farmers in this study sample use several technologies which may include yam mini-sett technology (Note: TE scores are relative measures that track efficiency relative to the most efficient observations within a particular sample. Thus, a caveat is that any difference between this study's estimates and previous works could simply be reflecting differences in heterogeneity across samples).

Accounting for ecology-specific production technologies shows that the mean yam MTE is 0.33. Like the TE, the MTE for yam is also constant across the study period but improves from

southern to northern Ghana. Specifically, the most technically efficient yam farmers relative to the industrial production frontier are those in the Sudan Savannah zone, with an MTE of 0.55. Fig 2 shows that Sudan Savannah farmers are followed by their peers in the Guinea Savannah (0.43), the Transition (0.34), and then the Forest (0.24) zones. Even though there is a technological gap of 28%, the overall results for yam show that yam MTE could be enhanced by improving TE.

**4.3.3 Cocoyam production.** The mean TGR for cocoyam, estimated at 0.80, indicates that the ecological technology gap is about 20%. The temporal trend from Fig 1, panel a3 shows that except for dips in 1991/1992 and again in 2011/2012, the TGR for cocoyam has remained close to its mean value of 0.80 over the study period. This mimics the yam TGR dynamics, but at a lower magnitude. At the ecology level, the TGR for cocoyam is higher in the Forest zone (0.82) than in the Transition zone (0.77).

Ecology-specific TE for cocoyam is estimated at a mean of 0.47 across the entire sample. Unlike the TGR, the cocoyam TE has been declining at a rate of about 5% from 1987 to 2017 (Fig 1, panel b3). Contrary to the TGR, cocoyam TE is higher in the Transition (0.56) than in the Forest (0.48) zone, as depicted in Fig 2. The dynamics in TGR and TE culminate in an estimated mean MTE of 0.36 for cocoyam. This follows the temporal trend exhibited by TE but improves from northern to southern Ghana. The overall results for cocoyam show that production could be enhanced by improving TE.

**4.3.4 Plantain production.** The mean TGR for plantain (0.85) is second only to cassava among the starchy staples considered. This indicates a plantain technology gap of about 15%. Fig 1, panel a4 shows that the TGR for plantain peaked in the period between 2004/05 and 2012/13 and has generally increased by about 3% annually from 0.72 in 1987 to 0.85 in 2017. At the ecology level, plantain TGR decreases from Transition zone (0.94), and Forest zone (0.87) to the Coastal Savannah zone (0.75). Thus, plantain farmers in the Transition zone are closest to the meta-frontier, while those in the Coastal Savannah zone can increase their productivity by about 25% if they adopt the best available technology.

Ecology-specific TE for plantain is estimated at a mean of 0.54 across the entire sample. The TE for plantain has remained constant over the study period (Fig 1, panel b4). The ecological zone ranking (high to low) of plantain TE is from Coastal Savannah (0.68), Forest (0.58), to the Transition (0.50) zones. In terms of MTE, the mean for plantain is 0.49. The MTE ranking (high to low) for plantain is from Forest (0.51) and Coastal Savannah (0.51) to the Transition zone (0.47). Plantain production could be enhanced by improving TE. The low MTE estimates for plantain production in this study are indicative of what has been reported in the literature. For instance, MoFA [5] indicated that actual average plantain yields represent only 32% of what is achievable; even the best-performing district attains less than 50% of the estimated potential yield.

## 4.4 Which factors affect starchy staple technical inefficiency and technology gaps in Ghana?

S7 Table in S1 File reports the estimates of factors that influence TE and TGR across crops and ecologies. Results for the case of TE are captured by the ecology frontiers and that of the TGR is captured by the meta-frontier. With TE, negative [positive] values indicate that the variables reduce [increase] technical inefficiency. For the case of TGR, negative [positive] values indicate that the variables reduce [increase] the technology gap.

The following caveats are to be noted in interpreting results from the effects of farmer-specific and institutional variables on TE and TGR presented in S7 Table in S1 File. First, these variables were harmonized across different surveys and used mainly as controls in estimating

the drivers of differences in starchy staples production outcomes in Ghana. Due to this level of aggregation, we assume only modest exogenous variation in the control variables and therefore do not claim causality. Notwithstanding, given this study's focus on establishing which of technological gaps or technical inefficiencies contribute more to production shortfalls, we believe the outcomes of these farmer characteristics and institutional factors can help to direct policies by examining their qualitative association with the outcomes of interest. To this end, we discuss these variables in terms of general associations and not in terms of individual coefficients.

**4.4.1 Farmer characteristics.** Except for yam production in the Sudan Savannah zone, male farmers report higher TE scores than female farmers across crops and ecologies, but sex had no significant effect on the TE of plantain production. Also, the effect of sex on TGR was only limited to yam production where the results showed a technology gap of 0.47 between males and females. These results show that women are doubly disadvantaged in starchy staples production as they are both less technically efficient and have higher technology gaps when compared to their male counterparts. These phenomena are supported by literature for rice [10], yam [24], and vegetables [23]. Results in S7 Table in S1 File show that age reduces inefficiencies in starchy staple production. At the national level, these effects are more pronounced in yam production, modest for cassava and plantain, and nonexistent for cocoyam. Thus, age has generally positive associations with TE and technological gaps. The nonlinear effect of age is only present for cassava production. Older farmers tend to be more inefficient in starchy staples production and exhibit wider gaps in adopting modern technologies. Owusu Asante et al. [24] found similar effects of age on yam farmers' technical inefficiency in Ghana's Ashanti and Brong-Ahafo regions. Education also showed non-linear effects only for yam production efficiency but not for the other starchy staples. Mostly, education was negatively associated with technical efficiency and adoption of superior technologies for yam production.

**4.4.2 Institutional factors.** Where statistically significant, general results for land ownership show that farm households that own lands are more technically efficient than those farming on rented or communal lands. Land tenure security negatively influences farmers' technical inefficiency for all crops and ecologies except cocoyam farmers in the Transition zone. This highlights the positive effect of tenure security: we would expect landowners to be less risk averse and more incentivized to invest in management practices that improve soil fertility and land structure, and that eventually would lead to more efficient crop production than non-owners of land [10]. On the other hand, the effect of ownership on bridging technology gaps appears inconclusive. On its own, land ownership may not be an enough incentive to adopt and maintain the application of modern technologies. Modern production technologies may require significant extra labor and capital investment to apply. In such instances, resource-constrained farmers may rather opt for conservative land management techniques that assure a satisfactory level of output.

Access to agricultural extension services often plays a key role in helping farmers improve their crop production by promoting good agronomic practices and the adoption of better technology that improves efficiency. Our results confirm this *a priori* expectation by their generally negative associations with TE and TGR. Access to extension services reduced both technical inefficiency and technology gaps of yam and plantain production, but these effects were insignificant for cassava and cocoyam. As in this study, Owusu Asante et al. [24] also found that extension access reduced production inefficiency. Similarly, Egyir et al. [38] found that access to extension services increased plantain farmers' adoption of agro-chemicals which ultimately improved yields.

Access to credit is expected to enable farmers to acquire supporting inputs necessary to increase efficiency and productivity. The results in S7 Table in S1 File show generally positive associations with both TE and TGR. Access to production credit increased both technical

inefficiencies and technology gaps for cassava and yam production, but reduced technology gaps for plantain production. This counterintuitive finding on the negative effect of credit on TE and TGR is difficult to explain from the available data, but a possible prediction relates to the exact use of the accessed credit amounts. Do farmers invest credit funds in their starchy staple production? Oftentimes, farmers use the credit for other purposes than farm investment [39].

Access to mechanized services showed heterogeneous effects on starchy staple production. While mechanized services increased the technology gaps for plantain and cassava, it reduced technical inefficiency and technology gaps for yam production. This mixed result is not surprising and may be reflective of the heterogenous access to mechanized services across production ecologies and among farmers in Ghana [40]. That said, we are mindful of claiming too much inference to this finding given the conditions.

## 5. Discussion

The literature identifies technical inefficiencies and technology gaps as the main causes of low productivity in smallholder agricultural systems. Our discussion is tilted towards addressing the main question of the paper, which is to ascertain whether starchy staples production shortfalls in Ghana are predominantly driven by technical inefficiencies, technological gaps or both. As indicated, the technological gap depends on farmers' access to and adoption of the best production technology and shows how close the production technology in an ecology is to the best technology set. On the other hand, technical efficiency is driven by the ability and willingness of the farmer to properly use and combine the available input sets. Thus, management practices and enabling environment as opposed to new technologies, are key determinants of technical efficiency.

The results in Table 4 show that for cassava, yam, and plantain, agroecological technologies in northern Ghana tend to be closer to the meta-frontier technology indicating low technological gaps; however, the technology gaps increase consistently as one traverses southwards. Unlike the technological gaps, technical inefficiency scores for the various agro-ecologies did not seem to follow any pattern across space or time. Acheampong et al. [41] examined cocoyam farmers' rates of adopting improved technologies and found that farmers in the Transition zone had higher adoption rates than those in the Forest zone. Their results further showed that the high adoption of complementary production techniques like appropriate planting distance, timely weed control, and weedicide application in the Transition zone improved the TGR for that zone.

The MTE provides a basis for comparing the efficiency of starchy staple production for all agro-ecologies relative to the best available technology. The key observation from these results is that the technical inefficiency effects across all ecologies, as well as at the national level, outweigh the technological gap effects. Thus, even though both technical inefficiency and technological gaps exist across all ecologies, technical inefficiencies are always higher than technological gaps. These results suggest more room for improving the efficiency of producing all crops in all agro-ecologies. For example, while cassava production in the Guinea Savannah zone is the most efficient relative to the meta-frontier, production levels can still be increased by about 40% if the inefficiency effects are eliminated, given the prevailing input levels and production technologies. What could be the possible explanation for the trends we observe, where technical inefficiency effects tend to outweigh the technological gaps?

Over the years, the interest of policies in many developing countries, including Ghana, has been to improve production levels mainly through the introduction of new technologies or input subsidies [42, 43] while less attention is given to mechanisms that can increase the efficiency of production. For instance, while Ghana cocoa sector has benefited from several

fertilizer subsidy schemes, it is well documented that inappropriate use of the subsidized fertilizer in terms of timing and application rates was a common production problem [44]. It has also been found that providing subsidized fertilizer in addition to training on best management practices via farmer field schools results in relatively more benefits in terms of increased yields than providing subsidized fertilizer alone [45]. These observations suggest that increasing farmer access to improved/new technology (improved crop varieties, fertilizer, etc..,) solves only one part of the problem: human capital investments and training on proper technology usage are also needed. However, such add-on outreach activities often are not the focus of most programs or they get sidelined during the implementation stages of government policies (as was the case of the Root and Tuber Improvement Program [RTIP] (1999–2004)] and its successor, Root and Tuber Improvement and Marketing Program [RTIMP] (2005–2015) [46, 47]. The focus of the RTIP was on the multiplication and distribution of improved planting materials and integrated pest management, with less attention given to complementary mechanisms that increase technical efficiency.

A closer look at the performances of individual starchy staples reinforces the roles of policies and complementary inputs in efficient crop production. From the MTE scores, the order of best performing starchy staples were cassava, plantain, cocoyam, and yam. The higher efficiency scores for cassava might be attributed to the prominent attention it received in policies and programs such as the President's Special Initiative (PSI) on agribusiness and cassava (2002–2004), the RTIP and RTIMP, as well as the West Africa Agricultural Productivity Program [WAAPP] (2007–2017) that aimed at increasing productivity of starchy staples in Ghana.

The finding that yam is the most inefficiently produced starchy staple in Ghana is worrying for several reasons. Ghana is the world's second-largest producer of yam (behind Nigeria) but the leading exporter to the global market [48, 49]. Given that about 94% of yam produced in Ghana is destined for the international market, addressing the causes of production shortfalls has the potential to improve the livelihoods of actors in the yam value chain. At the local levels, yam holds keystone positions in the cultural expression of several tribes in Ghana [50, 51] and remains an essential food security crop. For instance, yam consumption per capita stood at 125kg in 2015 [49]; however, about 40% of yam harvested in the Guinea Savannah zone (the largest yam producing area) is lost through poor postharvest handling [48]. This, coupled with resource use inefficiencies, threatens the capacity of yam to contribute to achieving Ghana's SDG 2 targets.

The finding that persistent technical inefficiency is relatively more of a challenge to productivity improvement than technology gaps across all starchy staples considered in this study means that critical attention must be given to the enabling factors that foster productivity growth rather than searching for new technologies all the time. Already, smallholder farmers are risk averse, which affect their production decisions and adoption of new technologies [52]. Therefore, continuously exposing these farmers to new technologies without first improving the enabling environment that foster their adoption may be counterproductive. Studies have found that management and institutional variables such as extension services are important drivers of technology adoption and productivity [38, 53]. Our results in S7 Table in S1 File support this assertion, as we found that both TE and TGR for some starchy staples are significantly affected by access to extension services.

## 6. Conclusions

In developing countries, many farmers unwittingly waste significant productive resources through production and technological inefficiencies which entrench poverty and food

insecurity. Changing the status quo and fostering rapid economic transformation would require sustained improvements in agricultural productivity. In this study, we take advantage of a large Ghanaian dataset that spans over three decades and examine the spatiotemporal trends of production gaps of one of the most important staple crop categories, the starchy staples. Unlike similar studies thus far, and in addition to the unique dataset used, the method adopted accounts for technological heterogeneity across the main ecological zones in Ghana where these starchy staple crops are cultivated. Using the stochastic meta-frontier approach, technical efficiency, meta-technical efficiency, and technology gap ratios are measured across crops and ecological zones. This analysis enables us to provide a broad understanding of the major sources of production shortfalls within the starchy staples industry of Ghana.

The key findings suggest that there are significant shortfalls in starchy staples production, which could be attributed to pure farmer technical inefficiencies and less to technology gaps along agroecological lines. The results show that the current set of production technologies used by starchy staple farmers in Ghana across ecologies are quite similar as the ecological technology gap that exists is about 20% for all crops except yam which has a technology gap of 35%. On the other hand, there exist significant production shortfalls that could be attributed to pure farmer technical inefficiencies. Respectively, about 35, 46, 53, and 45% of cassava, yam, cocoyam, and plantain production is lost due solely to technical inefficiencies and these have been relatively constant or increasing from 1987 to 2017. These trends suggest that mechanisms that improve the efficiency of resource use among starchy staple farmers have not received the needed attention over time. With the large production shortfalls attributable mostly to technical inefficiency, one implication of our results is that the starchy staple crop sector could benefit from improvements in conditions that enhance farmer technical efficiencies, such as providing starchy staple farmers more access to complementary production inputs such as extension services and better agronomic practices.

Further, the results also show that for any given starchy staple crop, some ecological zones perform better than others, thereby indicating heterogeneity in both technology and production management practices. We generally find that across all the crops considered, the technology gap across ecological lines is minimal but pure farmer technical inefficiency is chronic. This means that the production shortfalls for starchy staples in Ghana could be reduced if the causes of pure farmer technical inefficiency are adequately addressed. Existing policies aimed at improving the technology level (e.g., those supplying improved planting materials and subsidized inputs) could also emphasize the efficient use of these technologies. Stakeholders could learn important lessons from the frontier enhancing technologies and practices in high-performing ecological zones, improve them further and disseminate such improved technologies to less-performing agro-ecologies, guided, of course, by contextual factors.

## Supporting information

**S1 File.**
(DOCX)

## Acknowledgments

The authors gratefully acknowledge the Ghana Statistical Service, Institute of Statistical Social and Economic Research (ISSER), Economic Growth Center (EGC), and the International Food Policy Research Institute (IFPRI) for making the dataset available for the study. All errors are the responsibility of the authors. The findings and conclusions in this publication are those of the authors and should not be construed to represent any official USDA or U.S.

Government determination or policy. This research was supported in part by the U.S. Department of Agriculture, Economic Research Service.

## Author Contributions

**Conceptualization:** Francis Tsiboe.

**Data curation:** Francis Tsiboe.

**Formal analysis:** Francis Tsiboe.

**Methodology:** Francis Tsiboe.

**Supervision:** Isaac Gershon Kodwo Ansah.

**Writing – original draft:** Isaac Gershon Kodwo Ansah, Mark Appiah-Twumasi.

**Writing – review & editing:** Isaac Gershon Kodwo Ansah, Mark Appiah-Twumasi, Francis Tsiboe.

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
