## [Decision Letter · Decision Letter 0]

31 Oct 2022

PONE-D-22-23407Starchy staples production shortfalls in Ghana: Technical inefficiency effects outweigh technological differences across ecologiesPLOS ONE

Dear Dr. Ansah,

Thank you for submitting your manuscript to PLOS ONE. After careful consideration, we feel that it has merit but does not fully meet PLOS ONE’s publication criteria as it currently stands. Therefore, we invite you to submit a revised version of the manuscript that addresses the points raised during the review process.

We look forward to receiving your revised manuscript.

Kind regards,

Muhammad Tayyab Sohail

Academic Editor

PLOS ONE

Journal Requirements:

3. We note that Figure 1 in your submission contain map images which may be copyrighted. All PLOS content is published under the Creative Commons Attribution License (CC BY 4.0), which means that the manuscript, images, and Supporting Information files will be freely available online, and any third party is permitted to access, download, copy, distribute, and use these materials in any way, even commercially, with proper attribution. For these reasons, we cannot publish previously copyrighted maps or satellite images created using proprietary data, such as Google software (Google Maps, Street View, and Earth). For more information, see our copyright guidelines: http://journals.plos.org/plosone/s/licenses-and-copyright.

Reviewers' comments:

Reviewer's Responses to Questions

**Comments to the Author**

1. Is the manuscript technically sound, and do the data support the conclusions?

Reviewer #1: Yes

Reviewer #2: Partly

2. Has the statistical analysis been performed appropriately and rigorously? 

Reviewer #1: Yes

Reviewer #2: Yes

3. Have the authors made all data underlying the findings in their manuscript fully available?

Reviewer #1: Yes

Reviewer #2: Yes

4. Is the manuscript presented in an intelligible fashion and written in standard English?

Reviewer #1: Yes

Reviewer #2: Yes

5. Review Comments to the Author

Reviewer #1: Comments for the paper

“Starchy staples production shortfalls in Ghana: Technical inefficiency effects outweigh technological differences across ecologies”

The paper highlights an important topic in agricultural technology literature. The authors assess the production shortfalls of a selected number of major starchy staple crops in Ghana. They utilize meta-frontier efficiency analysis using secondary data obtained from the national socio-economic panel data sets.

An interesting finding of the study is that the biggest proportion of production shortfall is influenced by farmer technical inefficiencies rather than technology gaps. This is an interesting finding as it questions the assumption of farmers’ rationality in the adoption and utilization of agricultural technologies to enhance production and productivity. The recommendation is that farmers' managerial skills and efficient use of existing technologies is very critical for enhanced adoption and impact of agricultural innovations. I enjoyed reading this paper.

The paper is well written and therefore has very few comments.

Comments:

1. Datasets: the authors mentioned in the introduction that the dataset was pooled, but they indicated having a panel dataset in the methodology section. Panel data enhances the efficiency of the results and hence confidence in the generalization of the results; they should make this clear

2. Comparison of production shortfalls across different agro-ecological zones: I probably missed how the authors controlled for differences of various agro-ecologies favoring different crops, i.e. production differences should be compared across similar agroecological zones and using similar technologies

3. Consider adding a paragraph or two in the conclusions section highlighting the key shortcomings of the study (for future research) and key policy recommendations generated from the study.

Reviewer #2: 1. "Staple crop production in developing Africa provides both food and income to participating households, hence closing the yield gaps is crucial to improving food security and nutrition" - This is not important in the abstract

2. Be specific on the FAO reports in line 47

3. The reason underlining your study is weak. A better justification would do. This reason provided is different from the main objective of the paper.

4."The TE estimates showed that male farmers were marginally more efficient because of higher seeding and fertilizer application rates, while the MTE and TGR showed that female-managed farms were closer to the MSF than that of their male counterparts" How was this estimated? Were they considering two variables or same? Revisit the statement

5. "Furthermore, Tsiboe et al. (2019) showed that ecological gaps are very small for pepper, modest for tomato, and high for okra" - What is the main difference between this paper and your current study? Is it only about the crops?

6. "Given these scales of production, yields were estimated at 0.66, 0.96, 0.31,

and 0.53 Mt/ha for cassava, yam, cocoyam, and plantain, respectively. Except for cocoyam, these yields also vary along ecological lines" - These figures should be compared with MoFA standards

7. Your Tables did not tell your readers how the dependent variable in model 6 was captured. Is it in physical or value terms? Since your work is on four starch foods, using physical measure could be misleading. On the other hand, using value terms should be done with care due to inflation differentials.

8. "land contributes the highest returns to yam production in all ecological zones" - This statement is very misleading in this paper. You have your output and input variables in the frontier model normalized by land (ha) and you went ahead to include land in the model? This cannot be correct

9. Some diagnostic findings of model fitness and data adequacy should be conducted and mentioned in the discussions and in the conclusion.

10. "The key findings suggest that there are significant shortfalls in starchy staples production among agroecological zones, which could be more attributed to technical efficiencies and less to technology gaps" - Since the efficiencies determines the levels of individuals technologies and the technology gaps measures the distance to the potential technology, it becomes difficult to compare. Authors should really justify this comparison.

11. Authors should concentrate on their findings to conclude and not quote the works of others.

12. The essence of using the meta frontier model is to compare efficiencies for different technologies. If authors went ahead to estimate ecological gaps, then comparison be clear on technological lines and ecological lines. Use the meta efficiency scores to do this rather than the individual efficiency scores and the gaps.

6. PLOS authors have the option to publish the peer review history of their article (what does this mean?). If published, this will include your full peer review and any attached files.

Reviewer #1: No

Reviewer #2: No

---

## [Author Response · Author response to Decision Letter 0]

6 Dec 2022

Dear Editor, 

We have attached a response to reviewers file. 

Thank you.

---

## [Decision Letter · Decision Letter 1]

3 Feb 2023

PONE-D-22-23407R1Starchy staples production shortfalls in Ghana: Technical inefficiency effects outweigh technological differences across ecologiesPLOS ONE

Dear Dr. Ansah,

Thank you for submitting your manuscript to PLOS ONE. After careful consideration, we feel that it has merit but does not fully meet PLOS ONE’s publication criteria as it currently stands. Therefore, we invite you to submit a revised version of the manuscript that addresses the points raised during the review process. Please submit your revised manuscript by Mar 20 2023 11:59PM. If you will need more time than this to complete your revisions, please reply to this message or contact the journal office at plosone@plos.org. Please include the following items when submitting your revised manuscript:A rebuttal letter that responds to each point raised by the academic editor and reviewer(s). You should upload this letter as a separate file labeled 'Response to Reviewers'.A marked-up copy of your manuscript that highlights changes made to the original version. You should upload this as a separate file labeled 'Revised Manuscript with Track Changes'.An unmarked version of your revised paper without tracked changes. You should upload this as a separate file labeled 'Manuscript'.

We look forward to receiving your revised manuscript.

Kind regards,

Muhammad Tayyab Sohail

Academic Editor

PLOS ONE

Reviewers' comments:

Reviewer's Responses to Questions

**Comments to the Author**

1. If the authors have adequately addressed your comments raised in a previous round of review and you feel that this manuscript is now acceptable for publication, you may indicate that here to bypass the “Comments to the Author” section, enter your conflict of interest statement in the “Confidential to Editor” section, and submit your "Accept" recommendation.

Reviewer #2: (No Response)

Reviewer #3: (No Response)

2. Is the manuscript technically sound, and do the data support the conclusions?

Reviewer #2: Partly

Reviewer #3: Yes

3. Has the statistical analysis been performed appropriately and rigorously? 

Reviewer #2: Yes

Reviewer #3: Yes

4. Have the authors made all data underlying the findings in their manuscript fully available?

Reviewer #2: No

Reviewer #3: Yes

5. Is the manuscript presented in an intelligible fashion and written in standard English?

Reviewer #2: Yes

Reviewer #3: Yes

6. Review Comments to the Author

Reviewer #2: 1. The key aim of the stochastic Meta-frontier technique is technological comparison. How was this addressed in the work.

2. Improvements in crop productivity by closing production gaps in rural economies are generally touted as an important pathway to achieving three main developmental goals, namely food security, reduced poverty, and sustainable resource utilization ==> This statement should be deleted in the conclusion because authors did not address that

3. Quote Onumah, Onumah, Al-Hassan, Bruemmer (2013) to support literature on Meta-frontier approach in the Cocoa industry

Reviewer #3: General Comment:

This study investigates yield gaps in starchy staples in Ghana based a meta-frontier efficiency analysis. Mainly, the authors use household survey data to investigates the yield gaps of the starchy staples across various agroecological zones in Ghana. Their analysis highlights differences in yield gaps across the agroecological zones. These differences are mainly associated with technical inefficiency than technological differences, which is an interesting finding. However, it is not clear what does technical inefficiency mean. Does it refer to technical deficiencies? It would have been great when the authors could have highlighted the key reasons behind the so-called inefficiency. In my view, this is the main shortcoming of the manuscript, other shortcomings are listed below. Thus, I suggest considering it for publication after addressing the shortcomings.

First, the abstract could be longer. Currently, it does not highlight the research gaps, method applied, and implications of the study.

Second, the introduction section needs to be improved, mainly be updating the references based on the recent studies. See the specific comments for the details.

Third, it would be better to separate the result and discussion section. The authors could highlight the key findings in the result section, with their immediate discussion. In the discussion section, they could highlight the key novelties of the studies, comparing their results with other studies, discussing the limitation and policy implications. Moreover, the authors could have highlighted the reasons behind the differences in the yields based on the survey data. Currently, these reasons are diluted in the huge text. Even these reasons are vaguely mentioned as technical inefficiency in the conclusion. It would be better for readers and policy makers to pinpoint these technical inefficiencies.

Specific comments:

L43-36: Please update these numbers for the recent FAO report in 2022, see https://www.fao.org/publications/sofi/2022/en/

L52-59: Please update these statements based on the recent FAO report in 2022, see https://www.fao.org/publications/sofa/2022/en/. Here, the authors may highlight the crop yield gaps in Africa, e.g., see Pradhan et al. 2015 (https://doi.org/10.1371/journal.pone.0129487)

L67: In stead of footnotes, these numbers could be included in the main text in a condensed form.

L68-77: This paragraph argument that inefficient use of resources as a cause of yield gaps without any citation. Often these gaps are due to lack of resources, e.g., water, seed, and fertilizer , e.g., see Ladha et al. 2020 (https://doi.org/10.1016/bs.agron.2020.05.006).

L81: May be “Owusu Asante and colleagues [19]” instead of [19] only. Better to mention the author names in such cases.

L198: Here, the authors could also refer to SDG 2.3 on doubling crop productivity, contribution to rescue SDG from failing, e.g., see Pradhan 2023 (https://doi.org/10.1093/nsr/nwad015).

L112-210: This is a very long section on literature. It is interesting to read. However, the added value of it is not clear. Instead, it may divert the reader’s attention from this study. Thus, it might be better to have very condense justification of the selected method in the method section.

L371: “in the interest of space” is not a valid argument here. The authors have used a lot of space to justify the methods which can be move to SI. The authors would present and exploit the results properly.

7. PLOS authors have the option to publish the peer review history of their article (what does this mean?). If published, this will include your full peer review and any attached files.

Reviewer #2: No

Reviewer #3: No

---

## [Author Response · Author response to Decision Letter 1]

20 Mar 2023

Please, w have provided a response file as an attachment to the submission.

Thank you.

---

## [Decision Letter · Decision Letter 2]

10 Apr 2023

Starchy staples production shortfalls in Ghana: Technical inefficiency effects outweigh technological differences across ecologies

PONE-D-22-23407R2

Dear Dr. Ansah,

We’re pleased to inform you that your manuscript has been judged scientifically suitable for publication and will be formally accepted for publication once it meets all outstanding technical requirements.

Kind regards,

Muhammad Tayyab Sohail

Academic Editor

PLOS ONE

Additional Editor Comments (optional):

Reviewers' comments:

Reviewer's Responses to Questions

**Comments to the Author**

1. If the authors have adequately addressed your comments raised in a previous round of review and you feel that this manuscript is now acceptable for publication, you may indicate that here to bypass the “Comments to the Author” section, enter your conflict of interest statement in the “Confidential to Editor” section, and submit your "Accept" recommendation.

Reviewer #2: All comments have been addressed

Reviewer #3: All comments have been addressed

2. Is the manuscript technically sound, and do the data support the conclusions?

Reviewer #2: Yes

Reviewer #3: Yes

3. Has the statistical analysis been performed appropriately and rigorously? 

Reviewer #2: Yes

Reviewer #3: I Don't Know

4. Have the authors made all data underlying the findings in their manuscript fully available?

Reviewer #2: Yes

Reviewer #3: Yes

5. Is the manuscript presented in an intelligible fashion and written in standard English?

Reviewer #2: Yes

Reviewer #3: Yes

6. Review Comments to the Author

Reviewer #2: This is my third time reading this paper. the authors have attempted to address my comments. I currently do not have any further comments.

Thank you.

Reviewer #3: Many thanks for addressing the comments. I am happy with the changes made. If possible, the authors may consider shortening the paper by avoiding the repetition, moving less important parts to supplementary, and removing a longer background at the starting of a section, e.g. L139-142, L307-316.

7. PLOS authors have the option to publish the peer review history of their article (what does this mean?). If published, this will include your full peer review and any attached files.

Reviewer #2: No

Reviewer #3: No

---

## [Editor Report · Acceptance letter]

12 Apr 2023

PONE-D-22-23407R2 

Starchy staples production shortfalls in Ghana: Technical inefficiency effects outweigh technological differences across ecologies 

Dear Dr. Ansah:

I'm pleased to inform you that your manuscript has been deemed suitable for publication in PLOS ONE. Congratulations! Your manuscript is now with our production department. 

Kind regards, 

on behalf of

Dr. Muhammad Tayyab Sohail 

Academic Editor

PLOS ONE